# Hyperbolic-Euclidean Deep Mutual Learning

Submission Id: 1814

## ABSTRACT

Graph neural networks (GNNs) exhibit powerful performance in handling graph data, with Euclidean and hyperbolic variants excelling in processing grid-based and hierarchical structures, respectively. However, existing methods focus on learning specific structures that are linked to the inherent properties of the underlying space, and fail to fully exploit their complementary properties in distinct geometric spaces, thereby limiting their ability to efficiently model and represent complex graph structures. In this paper, we propose a Hyperbolic-Euclidean Deep Mutual Learning (H-EDML) framework, which leverages the unique properties of hyperbolic space to effectively capture the hierarchical relationships present in graph data, while also utilizes the familiar Euclidean space to handle local interactions. Specifically, We design a topology mutual learning module to bolster the capacity of each single model to perceive the holistic topological structure of the graph. Then, we integrate a decision mutual learning module to further advance the models' comprehensive judgment capabilities towards graph data, thereby strengthening the robustness and generalization. Furthermore, we employ an attention-based probabilistic integration strategy for the final prediction to alleviate potential disparities in decision-making among different models. Extensive experiments on node classification are conducted on five real-world graph datasets and the results show that our proposed H-EDML achieves competitive performances compared to the state-of-the-art methods.

## KEYWORDS

Mutual learning, Graph neural network, Hyperbolic geometry, Euclidean geometry

## 1 INTRODUCTION

Graph data, a prevalent form of structured data, presents a rich and versatile framework for representing a wide range of complex real-world systems such as social networks, biological networks, citation networks, and recommendation systems [4, 33, 36, 48]. The inherent structure of graph data encapsulates rich relational information, offering a powerful representation that captures the complex interplay between entities and their interactions. Graph neural networks on Euclidean geometry (GNNs) and hyperbolic geometry (HGNNs) have proven to be instrumental in studying graph data and addressing a myriad of graph-related tasks [22].

At the core of GNNs lies the ability to learn and propagate information across nodes in a graph, enabling the model to capture the structural dependencies and contextual information crucial for understanding the underlying data [35]. By iteratively aggregating and updating node features based on their neighborhood connections, GNNs excel in capturing local and global patterns, making them well-suited for a wide range of graph-related tasks. With the rise of Graph Convolutional Network (GCN) [24], the capabilities and promise of GNNs have been prominently showcased, sparking a notable increase in related research [38, 40, 44].

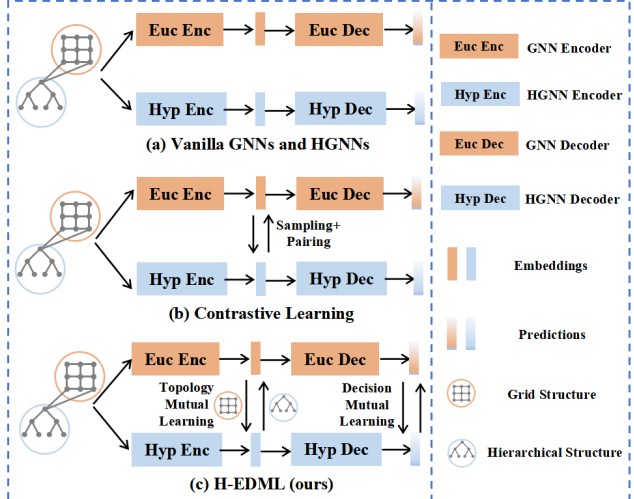

Figure 1: Diagram of our proposed H-EDML and existing graph learning approaches. (a) Vanilla GNNs and HGNNs; (b) Contrastive learning methods; (c) H-EDML. H-EDML combines the strengths of GNNs and HGNNs and is more comprehensive and efficient than contrastive learning methods.

In comparison to the polynomial growth of distances observed in Euclidean space, the exponential growth inherent to hyperbolic space provides a more appropriate framework for representing the hierarchical and tree-like data structures prevalent in graph data [13]. HGNNs capitalize on the intrinsic properties of hyperbolic geometry, allowing them to more effectively capture the relationships within hierarchical structures. As an advanced approach in the field of graph representation learning, HGNNs leverage hyperbolic geometry to not only enhance scalability but also improve the model's ability to differentiate between nodes with subtle topological differences [6, 12]. The diagram of vanilla GNNs and HGNNs are illustrated in Figure 1(a).

Due to the inherent limitations of the geometric spaces they operate in, both GNNs and HGNNs often fall short when it comes to handling the real-world complex networks. GNNs are constrained by the limitations of Euclidean geometry and show weaknesses in their ability to model large-scale graphs and capture the hierarchical structures prevalent in complex networks [6, 47]. On the other hand, while HGNNs leverage the hyperbolic space's negative curvature to effectively model hierarchical and tree-like structures, they struggle to handle uniform or flat graph structures due to the intrinsic properties of hyperbolic geometry [21].

Recognizing the complementarity between GNNs and HGNNs, researchers have advocated for contrastive learning methodologies to synergistically harness the respective advantages of both methods [20, 26, 46]. However, these contrastive learning approaches

introduce challenges due to the dependence on carefully curated positive and negative samples, as well as the complexities associated with subgraph sampling. Properly balancing and selecting these samples is crucial for ensuring effective contrastive learning, yet it requires significant computational resources and fine-tuning, which can limit the efficiency and scalability of these methods. The diagram of contrastive learning method is shown in Figure 1(b).

In this paper, we propose a novel approach termed Hyperbolic-Euclidean Deep Mutual Learning (H-EDML). Without necessitating meticulous selection of positive and negative samples as in contrastive learning methods, our approach amalgamates the strengths of both GNNs and HGNNs succinctly and efficiently. Through deep mutual learning, our model facilitates information exchange and structural interaction between hyperbolic and Euclidean spaces, enabling a synergistic learning process. Specifically, recognizing the distinct strengths of hyperbolic and Euclidean models in capturing diverse structural characteristics, we propose a topology mutual learning module to enhance the perceptual ability of each single model for the overall graph topology. Then, we introduce a decision mutual learning module aimed at enhancing the decision-making accuracy of the learning model by leveraging soft label information from the peer model. Furthermore, to address the potential residual discrepancies in decision-making between different models post mutual learning, we employ a probabilistic integration strategy based on an attention mechanism to obtain the ultimate prediction. The diagram of our H-EDML is illustrated in Figure 1(c). The main contributions of this paper are summarized as follows:

- We propose a novel framework, H-EDML, which addresses the issue of insufficient learning caused by the constraints of models relying solely on a single geometric space. Meanwhile, H-EDML achieves complementary information exchange through deep mutual learning between hyperbolic GNNs and Euclidean GNNs.
- We design two key modules in the H-EDML framework: topology mutual learning augments each single model's perception of different structures within a complex graph, and decision mutual learning enhances the decision-making capabilities of each model by leveraging comprehensive information interaction.
- We employ a two-stage training strategy to mitigate potential information redundancy and conflicts arising from the concurrent training of base models and attention network. Experimental results on five real-world graph datasets demonstrate that our H-EDML not only exhibits the highest stability but also achieves excellent classification performance, outperforming the state-of-the-art methods by up to 1.05%.

## 2 RELATED WORK

### 2.1 Graph Convolutional Networks

The field of graph convolutional networks (GCNs) [14] has witnessed significant growth and innovation in recent years, with a range of approaches being explored for effectively modeling and analyzing graph-structured data. The GCNs methods are commonly categorized into spectral and spatial methods, delineated by their respective mechanisms of information propagation and foundational principles of governing feature representation. Spectral methods

[2, 3, 5, 11, 24] use spectrum theory to extend the definition of convolution to graphs. Defferrard et al. [11] proposed to use Chebyshev polynomial as a filtering mechanism to reduce the time complexity of the convolutional kernel through polynomial approximation. In the realm of semi-supervised learning on graphs, Kipf et al. [24] simplified the Chebyshev network and proposed a first-order graph convolutional neural network. Bo et al. [3] proposed a Transformer-based set-to-set spectral filter along with learnable bases, effectively capturing both magnitudes and relative differences of all eigenvalues of the graph Laplacian. Spatial methods [7, 17, 28, 34, 50] principally revise node representations by incorporating features from both the nodes themselves and their adjacent nodes, thereby enabling the propagation and aggregation of information. Hamilton et al. [17] employed sampling to acquire a fixed number of neighbors for each node for information aggregation. Velivckovic et al. [38] integrated the attention mechanism into the graph convolutional network to weigh the importance of neighbor features during message passing. Zhang et al. [50] provided universal and efficient structure encoder and position encoder to enhance the structural learning capability of GNN architectures.

However, these methods concentrate on modeling within Euclidean Spaces, where traditional Euclidean frameworks frequently prove inadequate in encapsulating non-Euclidean characteristics present in real-world datasets, such as hierarchical structures.

### 2.2 Hyperbolic Graph Neural Networks

Hyperbolic graph neural networks (HGNNs) [47] have garnered significant utilization in graph data processing, primarily due to the exponential expansion characteristics of hyperbolic spaces aligning with the hierarchical structure frequently observed in graphs. HGCN [6] and HGNN [27] have both introduced extensions of graph neural networks into hyperbolic geometry. HGCN focused predominantly on tasks related to node classification and link prediction, while HGNN emphasized graph classification. Zhang et al. [51] introduced a graph attention network in the Poincaré ball model to embed hierarchical and scale-free graphs with minimal distortion. Bachmann et al. [1] proposed a theoretically grounded extension of GCN to constant curvature spaces. Despite the notable performance of hyperbolic GCNs, current hyperbolic graph operations do not strictly adhere to hyperbolic geometry, potentially limiting the efficacy of hyperbolic geometry and consequently impacting hyperbolic GCN performance. Zhang et al. [52] introduced a novel hyperbolic graph neural network ensuring strict adherence of learned node features to hyperbolic geometry. Dai et al. [10] proposed a convolutional network for hyperbolic-hyperbolic graphs directly operating on hyperbolic manifolds, thereby circumventing distortions stemming from tangent space approximations while preserving the global hyperbolic structure.

while both graph neural networks and hyperbolic graph neural networks focus on grasping a singular structure consistent with the underling space, they fall short in comprehensively capturing the intricate graph structure and extracting its features.

### 2.3 Deep Mutual Learning

Deep mutual learning (DML) [53] has emerged as a promising paradigm in the field of machine learning, enabling multiple neural

networks to collaborate and enhance model performance and generalization. Zhang et al. [53] proposed a sophisticated deep mutual learning strategy, facilitating collaborative learning among diverse networks throughout the training regimen. Zhao et al. [54] proposed a novel training approach for visual object tracking, based on mutual learning principles, to swiftly and effectively improve tracking performance. Wu et al. [43] designed a novel module for mutual learning as a foundational element in addressing significant object detection tasks, resulting in substantial improvements in detection efficacy. Zhao et al. [55] treated machine translation model and speech translation model as collaborative peers, effectively improving the performance of end-to-end speech translation. Xue et al. [45] pioneered a sophisticated methodology for deep adversarial mutual learning, utilizing domain-specific emotional cues to refine domain adaptive emotion classification. Li et al. [25] proposed a embedded fusion mutual learning model for the pathological image classification of the top three cancers.

Although DML has achieved commendable successes in various computer vision and natural language processing tasks, there is a scarcity of research that has explored its application in the realms of graphs and cross-geometry mutual learning.

## 3 PRELIMINARIES

A Riemannian manifold [31] $(\mathcal{M}, g)$ of dimension $n$ is a real and smooth manifold equipped with an inner product on tangent space $g_x : \mathcal{T}_x\mathcal{M} \times \mathcal{T}_x\mathcal{M} \to \mathbb{R}$ at each point $x \in \mathcal{M}$, where the tangent space $\mathcal{T}_x\mathcal{M}$ is a vector space and can be seen as a first order local approximation of $\mathcal{M}$ around point $x$. Hyperbolic space is a constant negative curvature Riemannian manifold equipped with a Riemannian metric. There are five isometric models of hyperbolic space, of which we work on the Poincaré ball model. An $n$-dimensional Poincaré ball model with curvature $-c(c > 0)$ is defined as $\mathbb{D}_c^n = \{x \in \mathbb{R}^n : c\|x\| < 1\}$ equipped with the Riemannian metric: $g_x^c = \lambda_x^2 g^E$, where $\lambda_x := \frac{2}{1-c\|x\|^2}$ and $g^E = \mathbf{I}_n$ is the Euclidean metric tensor.

Here we only give the definition of the operations necessary for a simple hyperbolic graph neural network [13]. The Möbius addition is defined as:

$$x \oplus_c y := \frac{(1 + 2c\langle x, y\rangle + c\|y\|^2)x + (1 - c\|x\|^2)y}{1 + 2c\langle x, y\rangle + c^2\|x\|^2\|y\|^2}, \quad (1)$$

where $x, y \in \mathbb{D}_c^n$. The Möbius scalar multiplication is defined as:

$$r \otimes_c x := (\frac{1}{\sqrt{c}})\tanh(r\tanh^{-1}(\sqrt{c}\|x\|))\frac{x}{\|x\|}, \quad (2)$$

where $x \in \mathbb{D}_c^n \backslash \{0\}$ and $r \in \mathbb{R}$. Similarly, the Möbius matrix-vector multiplication is defined as:

$$M \otimes_c x := (\frac{1}{\sqrt{c}})\tanh(\frac{\|Mx\|}{\|x\|}\tanh^{-1}(\sqrt{c}\|x\|))\frac{Mx}{\|Mx\|}, \quad (3)$$

where $x \in \mathbb{D}_c^n \backslash \{0\}$ and $M \in \mathbb{R}^{m \times n}$. During the construction of the hyperbolic graph neural network, a crucial step involves the conversion between the hyperbolic space and the tangent space, facilitated by the exponential map $\exp_x^c : \mathcal{T}_x\mathbb{D}_c^n \to \mathbb{D}_c^n$ and the logarithmic map $\log_x^c : \mathbb{D}_c^n \to \mathcal{T}_x\mathbb{D}_c^n$:

$$\exp_x^c(v) = x \oplus_c (\tanh(\sqrt{c}\frac{\lambda_x^c\|v\|}{2})\frac{v}{\sqrt{c}\|v\|}), \quad (4)$$

$$\log_x^c(y) = \frac{2}{\sqrt{c}\lambda_x^c}\tanh^{-1}(\sqrt{c}\| - x \oplus_c y\|)\frac{-x \oplus_c y}{\| - x \oplus_c y\|}, \quad (5)$$

where $x, y \in \mathbb{D}_c^n, x \neq y$ and $v \in \mathcal{T}_x\mathbb{D}_c^n\backslash\{0\}$. Ultimately, the definition of the hyperbolic non-linear activation incorporates the utilization of both exponential and logarithmic maps:

$$\sigma^c(x) = \exp_x^c(\sigma(\log_x^c(x))), \quad (6)$$

where $x \in \mathbb{D}_c^n$ and $\sigma$ is the non-linear activation in Euclidean space.

## 4 METHODOLOGY

### 4.1 Overview

Considering the complementarity of hyperbolic and Euclidean geometries in graph representation learning, and to overcome the constraints of contrastive learning dependent on positive and negative sample pairs, we propose a method termed Hyperbolic-Euclidean Deep Mutual Learning (H-EDML), which succinctly yet effectively facilitates collaborative learning between GNNs and HGNNs.

Specifically, our training procedure is bifurcated into two stages. In the first stage, we jointly train a GNN network and a HGNN network. Throughout this process, we introduce a topology mutual learning module to enhance the perceptual capabilities of each individual model regarding the overall graph topology. Additionally, we integrate a decision mutual learning module to leverage soft label information from the peer model to make more accurate prediction. Subsequently, in the second stage, we utilize the trained GNN and HGNN to train an attention network with an attention-based probabilistic integration strategy being implemented for the final prediction to mitigate potential discrepancies in decision-making across various models. The architecture is shown in Figure 2, and the training process of H-EDML is shown in Algorithm 1 in appendix. In the following subsections, we will give more details.

### 4.2 H-E Topology Mutual Learning

The significance of structural perception in graph learning is well recognized, with HGNN and GNN excelling in learning hierarchical and flat structures, respectively. In light of this, we introduce topology mutual learning (TML) mechanism to enhance the structural awareness of each network through topology structure interactions. As an illustration, HGNNs, proficient in learning hierarchical structures yet less effective in modeling graphs with low curvature or those closely resembling Euclidean spaces, can enhance their capacity for perceiving and modeling flat structures through the incorporation of the TML mechanism.

Let $\mathcal{G} = (\mathcal{V}, \mathcal{E})$ represent a graph, where $\mathcal{V}$ denotes the set of nodes and $\mathcal{E}$ denotes the set of edges. The feature matrix of nodes is denoted by $X \in R^{N \times d}$, where $N$ signifies the number of nodes and $d$ indicates the dimension of node features. Let $x_i$ denotes the feature representation of node $i$ and $y_i \in R^M$ denotes its one-hot class label, where $M$ is the number of classes. For node $i$, the embedding and output probability distribution of HGNN and GNN model are represented as $h_i^H, p_i^H$ and $h_i^E, p_i^E$, respectively.

We employ a topology perception module to acquire the structural attributes of diverse networks represented by a set of vectors $\{s_1, s_2, \cdots, s_N\}, s_i \in R^{N_{sample}}$. For small graphs, we encompass all nodes in our sampling strategy to comprehensively capture the

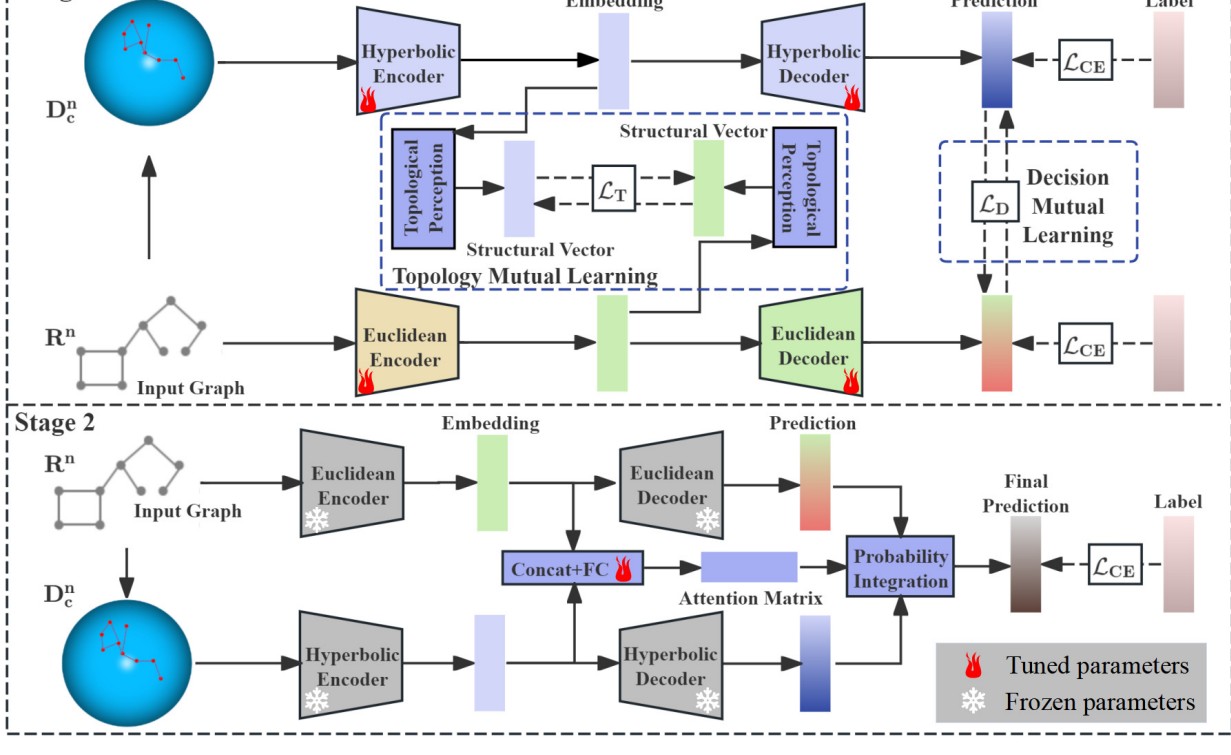

**Figure 2: The framework of H-EDML. In the first stage, the structural attributes of diverse network branches are characterized using a topological perception module. Subsequently, individual encoders and decoders are trained for each branch via topology mutual learning, decision mutual learning, and label supervision information. Moving to the second stage, the attention module is trained using the encoder and decoder trained in the first stage.**

structural interrelations within the graph with $N_{sample}$ signifying the total number of nodes in the graph. Conversely, for large graphs, we selectively sample the first and second-order neighbors surrounding the central node. This enables us to glean extensive insights into the structural relationships among nodes while maintaining computational feasibility within the constraints of larger graph sizes. In this case, $N_{sample}$ denotes the number of first and second-order neighbors around the central node. The elements of $s_i$ are computed as follows:

$$s_{ij} = \frac{e^{sim(h_i,h_j)}}{\sum_{j=1}^{N_{sample}} e^{sim(h_i,h_j)}}, \quad (7)$$

where $h_i$ represents the embedding of the network and $sim(h_i, h_j)$ measures the similarity between node embeddings. A conventional approach involves the direct utilization of Euclidean distance between node embeddings. However, the correspondence of node pairs in terms of similarity does not always align with their Euclidean distances [49]. To address this, we employ kernel function methodologies to articulate node similarities. Notable kernel functions encompass the linear kernel function, polynomial kernel function, radial basis kernel function, among others [19]. In this study, our preference lies with the polynomial kernel function:

$$sim(h_i^E, h_j^E) = K(h_i^E, h_j^E) = ((h_i^E)^T h_j^E + b)^d, \quad (8)$$

$$sim(h_i^H, h_j^H) = K(h_i^H, h_j^H) = ((\log_0^c h_i^H)^T \log_0^c h_j^H + b)^d, \quad (9)$$

where $b$, $c$ and $d$ are set to 0, 1 and 2.

Then, we can derive both the Euclidean structural attribute $s_i^E$ and hyperbolic structural attribute $s_i^H$ representing the structural distribution of each node $i$. Considering the inherent learning direction during the mutual learning process, we calculate the similarity of structural distributions across all nodes within the giving graph using the KL divergence. Consequently, we obtain the topology mutual learning loss:

$$\mathcal{L}_T^H = \sum_{i=1}^{N_{train}} KL(s_i^H \| s_i^E) = \sum_{i=1}^{N_{train}} \sum_{j=1}^{N} s_{ij}^H \frac{s_{ij}^H}{s_{ij}^E}, \quad (10)$$

$$\mathcal{L}_T^E = \sum_{i=1}^{N_{train}} KL(s_i^E \| s_i^H) = \sum_{i=1}^{N_{train}} \sum_{j=1}^{N} s_{ij}^E \frac{s_{ij}^E}{s_{ij}^H}, \quad (11)$$

where $N_{train}$ is the number of training nodes.

## 4.3 H-E Decision Mutual Learning

The limitations in achieving robust generalization stem from the disparate geometric spaces in which HGNNs and GNNs operate, impeding their ability to comprehensively capture and learn information about the entire graph. While TML mitigates the deficiencies

in model structure learning, but challenges remain in feature representation and decision making. To address this, we propose a decision mutual learning (DML) mechanism, which enables the acquisition of soft label information from a peer network. These soft labels provide additional insights that the network might have previously failed to capture. Specifically, we utilize the KL divergence to align the probability distributions of a network with its peer network and propose a decision mutual learning loss for each network to minimize the difference:

$$\mathcal{L}_D^H = \sum_{i=1}^{N_{train}} KL(p_i^H \| p_i^E) = \sum_{i=1}^{N_{train}} \sum_{j=1}^{M} p_{ij}^H \frac{p_{ij}^H}{p_{ij}^E}, \tag{12}$$

$$\mathcal{L}_D^E = \sum_{i=1}^{N_{train}} KL(p_i^E \| p_i^H) = \sum_{i=1}^{N_{train}} \sum_{j=1}^{M} p_{ij}^E \frac{p_{ij}^E}{p_{ij}^H}. \tag{13}$$

Each of the two networks not only benefits from mutual guidance at decision and structure levels but also receives supervision from the node label information:

$$\mathcal{L}_{CE}^H = \sum_{i=1}^{N_{train}} H(y_i, p_i^H) = -\sum_{i=1}^{N_{train}} \sum_{j=1}^{M} y_{ij} \log(p_{ij}^H), \tag{14}$$

$$\mathcal{L}_{CE}^E = \sum_{i=1}^{N_{train}} H(y_i, p_i^E) = -\sum_{i=1}^{N_{train}} \sum_{j=1}^{M} y_{ij} \log(p_{ij}^E). \tag{15}$$

Therefore, the total loss functions $\mathcal{L}^H$ and $\mathcal{L}^E$ for the hyperbolic graph neural network and the Euclidean graph neural network are formulated as follows:

$$\mathcal{L}^H = \mathcal{L}_{CE}^H + \alpha_1 \mathcal{L}_D^H + \beta_1 \mathcal{L}_T^H, \tag{16}$$

$$\mathcal{L}^E = \mathcal{L}_{CE}^E + \alpha_2 \mathcal{L}_D^E + \beta_2 \mathcal{L}_T^E, \tag{17}$$

where $\alpha_1$, $\beta_1$ and $\alpha_2$, $\beta_2$ are two pairs of hyperparameters used to balance the loss terms.

## 4.4 H-E Attention-based Probabilistic Integration

Utilizing decision mutual learning and topology mutual learning to facilitate the training of the two networks, we are able to derive two distinct probability distributions, denoted as $p_i^H$ and $p_i^E$, for any given node $i$. Due to variations in the capacities of distinct networks, our deep mutual learning approach, while mitigating this diversity, does not entirely eradicate it, leading to potential disparities in their decision-making. Therefore, to ascertain the ultimate probability distribution for node $i$, an attention-based probabilistic integration strategy is utilized to assign increased weights to the geometric probability distributions of greater significance.

We first concatenate the embeddings from the two networks and send them into fully connected layers to obtain an attention matrix $A$. Each of its rows is structured as follows:

$$A_i = S(\sigma((\delta([\log_0^c(h_i^H), h_i^E] W_1)) W_2)), \tag{18}$$

where $[\cdot, \cdot]$ denotes concatenation, $W_1$ and $W_2$ are two weight matrices, which can be implemented by two fully connected layers. $\delta$ and $\sigma$ correspond to activation functions that are implemented by ReLU and Sigmoid functions, respectively. The function $S$ denotes the softmax function applied along each row of the matrix. The attention matrix $A$ assigns two distinct weights to each node $i$:

$A_{i1}$ for the hyperbolic probability distribution and $A_{i2}$ for the Euclidean probability distribution. Subsequently, the final probability distribution for node $i$ can be derived from A:

$$p_i = A_{i1} p_i^H + A_{i2} p_i^E. \tag{19}$$

## 4.5 Two-stage Training Analysis

It is noteworthy that our H-EDML model introduces modifications to the conventional training process. Firstly, we employ an alternating training scheme to jointly optimize the two networks, ensuring gradual performance improvements for each network throughout this process. This significantly reduces training costs compared to separately training each network and then engaging in mutual supervision, with minimal performance loss. Secondly, our method adopts a two-stage training approach. Initially, the two networks undergo alternating training with assistance from decision mutual learning and topology mutual learning. Subsequently, the trained networks are utilized to train the attention mechanism. This strategy is implemented to mitigate the potential redundancy and conflicts in information that may arise if both graph neural networks and the attention mechanism are simultaneously trained in a single stage. Specifically, such conflicts could emerge from the influence of the output probability distributions of the peer network at the decision-making level during decision mutual learning and the computation of label-supervised loss using joint probability distributions. These factors could detrimentally impact the overall model performance. For a detailed description of the model training and optimization process, refer to Algorithm 1 in appendix. It is worth mentioning that in the second stage, we not only finalize the training of attention but also conclude the inference process.

## 5 EXPERIMENTS

In this section, we conduct extensive node classification experiments on five graph datasets and prove the superiority of our proposed H-EDML. In addition, we also conduct ablation studies to verify the effectiveness of our TML, DML and attention-based probabilistic integration. Finally, we analyze the impact of embedding dimensions, different underlying models on our H-EDML and a visualization result.

## 5.1 Experimental Settings

*5.1.1 Datasets.* We utilize five datasets in our experiments: Cora [33], Citeseer [15], Pubmed [29], Airport and Disease [6]. Cora, Citeseer and Pubmed are citation networks where nodes represent scientific papers, and edges are citations between them. Airport dataset describes the location of airports and airline networks, where nodes represent airports and edges represent airline routes. Disease dataset simulates the disease propagation tree, where node represents a state of being infected or not by SIR disease. We further compute $\delta-$hyperbolicity [16] to quantify the tree-likeliness of these datasets. A lower $\delta-$hyperbolicity denotes a more tree-like structure and $\delta = 0$ denotes a pure tree. The details of data statistics are shown in Table 1.

*5.1.2 Baselines.* We compare our method with the following state-of-the-art methods: (1) Neural network (NN) methods: deep neural networks that disregard graph topology, including MLP [30] and

**Table 1: Statistics of the experimental datasets.**

| Dataset | Nodes | Edges | Classes | Features | $\delta$ |
|---------|-------|-------|---------|----------|----------|
| Cora | 2,708 | 5,429 | 7 | 1,433 | 11 |
| Citeseer | 3,327 | 4,732 | 6 | 3,703 | 5 |
| Pubmed | 19,717 | 44,338 | 3 | 500 | 3.5 |
| Airport | 3,188 | 18,631 | 4 | 4 | 1 |
| Disease | 1,044 | 1,043 | 2 | 1,000 | 0 |

HNN [13]. (2) Euclidean graph neural network (Euclidean GNN) methods: deep neural networks that leverage both node embeddings and graph topologies in Euclidean spaces, including GCN [24], GraphSAGE [17], GAT [38], SGC [40], GraphCON [32], NodeFormer [41], SGFormer [42] and ACMP [39]. (3) Hyperbolic graph neural network (hyperbolic GNN) methods: approaches for modeling deep graph neural networks in hyperbolic spaces, including HGNN [27], HGCN [6], GIL [56], HGAT [51], LGCN [52], HGCL [26], F-HNN [8], H-GRAM [9] and LRN [18].

5.1.3 *Implementation Details.* In our experiments, we closely follow the parameter settings in [6] and obey the same dataset split for all baselines. We use standard splits in GCN [24] on citation network datasets, 70/15/15 percent splits for training, validation and test on Airport, and 30/10/60 percent splits on Disease. All methods use the following training strategy, including the same random seeds for initialization and the same early stopping on validation set with 200 patience epochs. The grid search is performed over the following search space: Learning rate: [0.001, 0.005, 0.01, 0.02]; Dropout rate: [0.0, 0.1, 0.2, 0.3, 0.4, 0.5, 0.6]; weight decay: [0, 1e-4, 5e-4, 1e-3]; Number of hidden layers: [1, 2, 3]. The results are reported over 10 random parameter initializations. We set the dimension of latent representation of all methods as 16 and the curvature for hyperbolic GNNs as 1. We designate HGAT and GCN as the fixed hyperbolic and Euclidean base models in our H-EDML framework. We optimize H-EDML with Adam [23]. All the experiments are conducted on NVIDIA GeForce RTX 3090 GPU and are implemented in Python using the PyTorch framework.

## 5.2 Experimental Results

**H-EDML combines the advantages of Euclidean GNNs and hyperbolic GNNs to achieve satisfactory performance.** In Table 2, we report the mean accuracy for three citation network datasets and the mean F1 score for the Airport and Disease datasets of our proposed methods and all baselines. Our H-EDML achieves the best performance across the other four datasets except for the Disease dataset with up to 1.05% improvement over the sub-optimal method, showcasing the strength of our approach. Notably, our method also secures the third-best performance on the Disease dataset, following only HGCL, a contrastive learning method and LRN which is specifically designed for highly hyperbolic data. However, our H-EDML model achieves a performance metric of 0.9253, which is marginally lower compared to the leading HGCL model with a score of 0.9342. Furthermore, it is evident that our proposed method has demonstrated significant advancements over our base models, HGAT and GCN, indicating the appropriateness of our

method in integrating heterogeneous space networks. Overall, our approach integrates the benefits of Euclidean GNN and hyperbolic GNN methods via decision mutual learning and topology mutual learning, yielding satisfactory results across all datasets.

**Euclidean GNNs are good at learning flat structures and hyperbolic GNNs are good at learning hierarchical structures.** In Table 2, the traditional NN methods perform poorly on all datasets, demonstrating the significance of graph topology in graph representation learning. Moreover, on the three citation network datasets with high $\delta$−hyperbolicity, Euclidean GNN methods outperform hyperbolic GNN methods overall, with the top-performing methods, aside from our own, belonging to the Euclidean GNN category. This suggests that Euclidean GNN methods are better able to capture and learn flat non-hierarchical structures. Similarly, on the Airport and Disease datasets with low $\delta$−hyperbolicity, hyperbolic GNN methods outperform Euclidean GNN methods overall, with the best and second-best methods falling within the hyperbolic GNN category. This further validates the advantages of hyperbolic networks in learning hierarchical structures.

## 5.3 Analysis

5.3.1 *Ablation Study.* To assess the impact of our proposed primary module and the two-stage training strategy, we conduct a series of ablation experiments, where 'w/o' denotes the removal of the corresponding component from our H-EDML framework: TML for topology mutual learning, DML for decision mutual learning, and ATT for the attention module. We achieve the removal of ATT from our method by assigning equal weights of 0.5 to both networks. The 'Single' denotes the use of single-stage training in our H-EDML framework. Analysis of the experimental results highlights the following observations in Table 3.

**Each component demonstrates its positive impact on the overall framework.** Firstly, the classification performance of our H-EDML exhibit enhancement subsequent to mutual learning on our base models, HGAT and GCN, underscoring the merit of integrating network interactions across disparate spatial domains. Subsequent removal of TML from our model leads to performance degradation, where the F1-score drops by 2.59% on the Airport dataset, emphasizing the pivotal role of our proposed topology mutual learning in facilitating enhanced comprehension of graph topology. In addition, the exclusion of DML results in an average performance decrease of 1.5% indicating the model's ability to acquire a more comprehensive understanding of node attributes and class distinctions through decision mutual learning. Furthermore, the performance decline in the absence of ATT highlights the varying significance of hyperbolic GNN and Euclidean GNN for distinct nodes within diverse graph structures. The performance of two-stage training is generally better than that of single-stage training, which further corroborates the previous analysis that single-stage training brings information redundancy and conflict. Ultimately, merging TML, DML, and ATT and employing two-stage training in our H-EDML model produces the best results, thus reaffirming the effectiveness of each component of our proposed framework as well as the two-stage training strategy.

5.3.2 *The Impact of Embedding Dimensions.* In order to verify whether our method can maintain the advantage of hyperbolic

**Table 2: Accuracy (%± standard deviation) for Cora, Citeseer and Pubmed datasets and F1 score (%± standard deviation) for Airport and Disease datasets for node classification. Performance scores averaged over ten runs. The best results are boldfaced while the runner-ups are underlined. $\delta$ refers to gromovs $\delta-$hyperbolicity.**

| | Method | Cora ($\delta = 11$) | Citeseer ($\delta = 5$) | Pubmed ($\delta = 3.5$) | Airport ($\delta = 1$) | Disease ($\delta = 0$) |
|---|---|---|---|---|---|---|
| NN | MLP (2009) | 50.66±1.35 | 58.58±0.83 | 73.54±0.30 | 70.12±0.55 | 31.69±2.65 |
| | HNN (2018) | 55.64±0.58 | 58.66±1.44 | 69.49±0.55 | 81.34±0.47 | 43.48±1.76 |
| Euclidean GNN | GCN (2017) | 82.02±0.45 | 70.24±0.85 | 78.45±0.36 | 83.19±1.76 | 79.73±1.88 |
| | GraphSAGE (2017) | 77.84±0.66 | 67.11±1.56 | 76.88±0.56 | 88.55±0.84 | 75.29±2.76 |
| | GAT (2018) | 82.72±0.69 | 71.45±1.15 | 77.23±0.51 | 86.10±1.39 | 80.25±0.54 |
| | SGC (2019) | 81.91±0.63 | 71.95±0.55 | 78.60±0.24 | 84.07±1.91 | 79.98±2.10 |
| | GraphCON (2022) | 82.54±1.25 | 71.29±0.96 | 78.66±1.23 | 79.25±2.51 | 86.74±2.53 |
| | NodeFormer (2022) | 82.34±0.85 | 72.47±1.27 | 79.13±1.10 | 82.66±0.69 | 79.84±1.33 |
| | SGFormer (2023) | 83.11±0.75 | 72.35±0.54 | 79.25±0.65 | 91.67±1.21 | 88.55±2.16 |
| | ACMP (2023) | 82.65±0.78 | 73.01±1.14 | 78.52±0.54 | 88.96±1.74 | 89.15±2.19 |
| Hyperbolic GNN | HGNN (2019) | 78.99±0.61 | 70.20±0.61 | 76.94±1.12 | 84.87±2.09 | 82.16±1.41 |
| | HGCN (2019) | 78.41±0.77 | 68.29±0.95 | 77.03±0.49 | 88.82±1.66 | 89.36±1.02 |
| | GIL (2020) | 82.49±0.77 | 71.21±0.93 | 77.35±0.56 | 91.21±0.72 | 89.69±0.93 |
| | HGAT (2021) | 79.71±0.95 | 69.41±0.78 | 75.56±0.67 | 89.25±0.96 | 90.43±1.12 |
| | LGCN (2021) | 78.93±0.79 | 68.59±0.64 | 78.08±0.65 | 88.53±1.26 | 91.15±1.02 |
| | HGCL (2021) | 82.37±0.47 | 72.11±0.64 | 79.17±0.68 | 92.35±1.01 | **93.42±0.82** |
| | F-HNN (2022) | 81.04±1.78 | 71.12±0.66 | 77.66±1.04 | 91.85±0.86 | 92.03±1.21 |
| | H-GRAM (2023) | 81.56±1.58 | 72.16±1.66 | 78.05±1.84 | 89.89±2.16 | 91.22±2.54 |
| | LRN (2024) | 78.34±1.16 | 67.10±1.19 | 77.24±0.67 | 92.39±0.72 | 93.29±1.45 |
| | H-EDML (ours) | **84.07±0.61** | **73.96±0.22** | **79.68±0.38** | **93.44±0.63** | 92.53±0.72 |

**Table 3: The ablation experiment results where 'w/o' represents removal of the corresponding module and 'Single' denotes the use of single-stage training in H-EDML framework. The best results are boldfaced.**

| Method | Cora | Citeseer | Pubmed | Airport | Disease |
|---|---|---|---|---|---|
| GCN | 82.02 | 70.24 | 78.45 | 83.19 | 79.73 |
| HGAT | 79.71 | 69.41 | 75.56 | 89.25 | 90.43 |
| w/o TML | 83.01 | 73.13 | 78.88 | 90.85 | 91.18 |
| w/o DML | 82.57 | 72.46 | 78.87 | 91.83 | 90.56 |
| w/o ATT | 83.25 | 73.28 | 79.08 | 91.63 | 90.89 |
| Single | 82.74 | 72.81 | 79.06 | 91.47 | 91.15 |
| H-EDML | **84.07** | **73.96** | **79.68** | **93.44** | **92.53** |

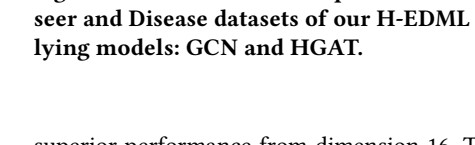

(a) Citeseer    (b) Disease

**Figure 3: The influence of representation dimension on Citeseer and Disease datasets of our H-EDML and the two underlying models: GCN and HGAT.**

graph neural networks with lower embedding dimensions to obtain efficient embedding representation after the integration of graph neural networks in Euclidian space, we conduct experiments by setting different embedding dimensions on the highly hierarchical Disease dataset and the weakly hierarchical Citeseer dataset. The experimental results are shown in Figure 3.

**H-EDML is robust to dimensions and consistently maintains good performance.** Initially, observations on the Disease dataset reveal a progressive enhancement in GCN performance with escalating embedding dimensions, albeit at a diminishing growth rate. HGAT stabilizes from dimension 8 onwards, while our H-EDML method mirrors this behavior, demonstrating consistent and

superior performance from dimension 16. This underscores the challenge of distortion inherent in graph neural networks operating on Euclidean geometry when dealing with highly hierarchical data. The inherent distortion that occurs in Euclidean space arises because the geometric properties of this space are not naturally suited to capturing the exponentially expanding structure of hierarchical or tree-like data. Although increasing the dimensionality of the Euclidean space can reduce some of this distortion, the fundamental misalignment between the flat geometry of Euclidean space and the complex, hierarchical structure of the data persists. Nevertheless, hyperbolic geometry-based graph neural networks effectively mitigate such distortions, yielding satisfactory embedding

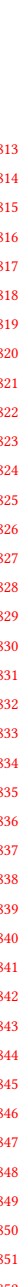

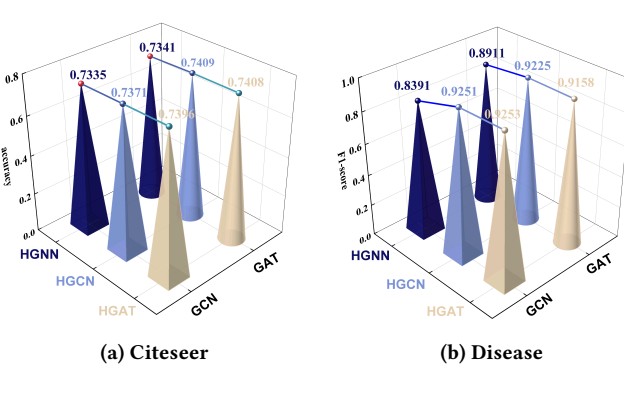

(a) Citeseer          (b) Disease

**Figure 4: The impact of different base model pairs on our proposed H-EDML framework. We select GCN and GAT as the Euclidean base models and HGNN, HGCN and HGAT as the hyperbolic base models.**

representations at lower dimensions owing to their intrinsic alignment with hierarchical structures. Moreover, compared to other methods, our H-EDML consistently maintains high stability and superior performance on the less hierarchical Citeseer dataset.

*5.3.3 The Impact of Different Underlying Models.* **H-EDML shows good adaptability.** To investigate whether our H-EDML framework depends on specific bace model pairings when handling different datasets, we conduct experiments employing various combinations of base models. As illustrated in Figure 4, it is apparent that, apart from HGNN, the other base model combinations consistently demonstrate strong performance across distinct dataset types, underscoring the practical effectiveness of our methodology. Consequently, when faced with diverse datasets, there is no imperative need to exhaust resources in exploring numerous combinations; instead, an initial random selection suffices. This further illustrates the adaptability of our H-EDML across diverse base models and flexibility for various datasets.

**H-EDML demonstrates strong robustness.** It can also be noticed from Figure 4 that our approach does not require high-performance standards from individual base models. For instance, on the Citeseer dataset, The classification accuracies of HGAT and GAT are only 69.41% and 71.45% respectively, but following our decision mutual learning and topology mutual learning processes, the classification accuracy improves to 74.08%. Similarly, on the Disease dataset, where the F1-score of HGCN and GAT are only 89.36% and 80.25% respectively, our method achieves a remarkable 92.25% post deep mutual learning. This shows that H-EDML can robustly achieve better results regardless of the performance of the base models, indicating that it is insensitive to the performance differences of the base model and demonstrates strong robustness.

*5.3.4 Visualization.* We present experimental evidence illustrating the efficacy of our approach in enhancing node representation learning within base models, GCN and HGAT, using the Airport dataset. We visualize the performance of these models before and after applying our mutual learning technique. To facilitate this comparison, we leverage the T-distributed Random Neighbor Embedding (t-SNE) tehcnique [37] to project the high-dimensional embeddings from

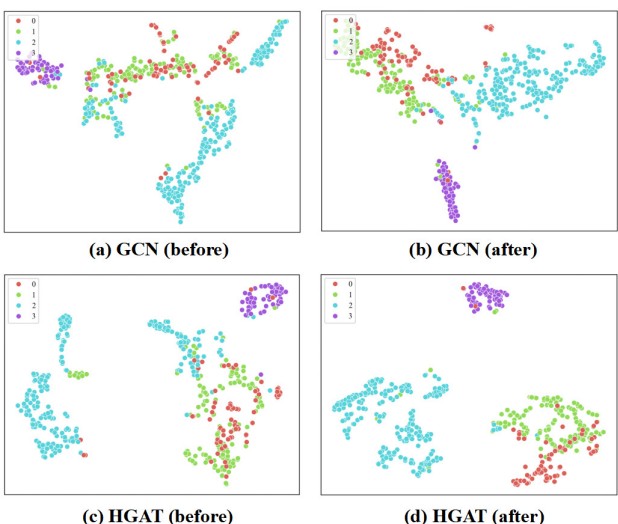

(a) GCN (before)          (b) GCN (after)

(c) HGAT (before)         (d) HGAT (after)

**Figure 5: Visualization of the Airport dataset. (a) Embedding of GCN before our deep mutual learning; (b) Embedding of GCN after our deep mutual learning; (c) Embedding of HGAT before our deep mutual learning; (d) Embedding of HGAT after our deep mutual learning.**

the final layer of each model onto a two-dimensional plane for visual analysis. We assign colors to each data point, with distinct colors denoting different classes.

**H-EDML enhances the separability of node embeddings.** The visualization results, as illustrated in Figure 5, demonstrate that following the implementation of H-EDML, the node embeddings learned by GCN and HGAT exhibit more distinct boundaries between various classes. For instance, the separability between classes 2 and 3 is notably improved, and the initial overlap between classes 0 and 1 is reduced. This suggests that the enhanced node embeddings, resulting from our mutual learning approach, more effectively capture the fundamental structure of the graph, better preserving class distinctions and thereby improving the overall separability of the node embeddings.

## 6 CONCLUSION

In this paper, we propose Hyperbolic-Euclidean Deep Mutual Learning, a two-stage training framework, which can fully excavate structural information and make more comprehensive decisions through the topology interaction and decision interaction of two networks. Specifically, we first design a topology mutual learning module to enhance each individual model's perception of the overall graph structure. Then, a decision mutual learning module is introduced in the decision level to obtain soft label information from the peer model, so as to make a more comprehensive decision. In addition, we introduce an attention-based probabilistic integration strategy to make final predictions to mitigate potential differences in decision-making between different models. Extensive experiments demonstrate the superiority of H-EDML, compared with the state-of-the-art methods.

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

## A ALGORITHM

---

**Algorithm 1** algorithm of H-EDML

---

**Input:** Graph $\mathcal{G} = (\mathcal{V}, \mathcal{E})$ with node labels $\mathcal{Y}$; Trainable hyperbolic GNN $g_H(\cdot)$; Trainable Euclidean GNN $g_E(\cdot)$; Trainable attention module $g_A(\cdot)$; Number of training epochs for hyperbolic and Euclidean GNN model $N_1$; Number of training epochs for attention $N_2$; Loss hyperparameters $\alpha_1, \beta_1, \alpha_2, \beta_2$

1: Parameter $\theta_H$, $\theta_E$ and $\theta_A$ initialization;

**Output:** Trained hyperbolic and Euclidene GNN model and attention module.

2: **for** $i = 1, \cdots, N_1$ **do**

3:     Learn node embedding $h^H$ and $h^E$ ;

4:     Calculate label distribution $p^H$ and $p^E$ using $h^H$ and $h^E$;

5:     Calculate topology mutual learning loss $\mathcal{L}_T^H$ and $\mathcal{L}_T^E \leftarrow$ Equation (10)(11);

6:     Calculate decision mutual learning loss $\mathcal{L}_D^H$ and $\mathcal{L}_D^E \leftarrow$ Equation (12)(13);

7:     Calculate label supervision loss $\mathcal{L}_{CE}^H$ and $\mathcal{L}_{CE}^E \leftarrow$ Equation (14)(15);

8:     Calculate overall classification loss $\mathcal{L}^H$ and $\mathcal{L}^E \leftarrow$ Equation (16)(17);

9:     Update hyperbolic GNN parameters $\theta_H \leftarrow \theta_H - \eta_1 \nabla \theta_H$;

10:     Update Euclidean GNN parameters $\theta_E \leftarrow \theta_E - \eta_2 \nabla \theta_E$.

11: **end for**

12: **for** $j = 1, \cdots, N_2$ **do**

13:     Calculate node label distribution $p \leftarrow$ Equation (19);

14:     Calculate classification loss using cross-entropy;

15:     Update Attention parameters $\theta_A \leftarrow \theta_A - \eta_3 \nabla \theta_A$.

16: **end for**

---

## B GRAPH NEURAL NETWORKS

Let $\mathcal{G} = (\mathcal{V}, \mathcal{E})$ represent a graph, where $\mathcal{V}$ denotes the set of nodes and $\mathcal{E}$ denotes the set of edges. The feature matrix of nodes is denoted by $X \in R^{N \times d}$, where $N$ signifies the number of nodes and $d$ indicates the dimension of node features. Let $x_i$ denotes the feature representation of node $i$ and $y_i \in R^M$ denotes its one-hot class label, where $M$ is the number of classes. For node $i$, the embedding and output probability distribution of HGNN and GNN model are represented as $h_i^H, p_i^H$ and $h_i^E, p_i^E$, respectively.

Let $\mathcal{N}(i) = \{j : (i, j) \in \mathcal{E}\}$ denote the set of neighbors of node $i \in \mathcal{V}$, $W^l$ and $b^l$ be weights and bias parameters for layer $l$, and $\sigma(\cdot)$ be a non-linear activation function. The message passing procedure in GNNs at layer $l$ for node $i$ can be summarized in the following three steps: feature transformation, neighborhood aggregation and non-linear activation:

$$x_i^{l,E} = W^l h_i^{l-1,E} + b^l, \quad (20)$$

$$z_i^{l,E} = x_i^{l,E} + \sum_{j \in \mathcal{N}(i)} \omega_{ij} x_j^{l,E}, \quad (21)$$

$$h_i^{l,E} = \sigma(z_i^{l,E}). \quad (22)$$

Analogous to the GNNs framework, the message propagation process in HGNNs at layer $l$ for node $i$ can be succinctly encapsulated by the following three steps: hyperbolic feature transformation, hyperbolic neighborhood aggregation and hyperbolic non-linear activation:

$$x_i^{l,H} = W^l \otimes_c h_i^{l-1,H} \oplus b^l, \quad (23)$$

$$z_i^{l,H} = x_i^{l,H} \oplus_c \exp_{x_i}^c \left( \sum_{j \in \mathcal{N}(i)} \omega_{ij} \log_{x_i}^c x_j^{l,H} \right), \quad (24)$$

$$h_i^{l,H} = \sigma^c(z_i^{l,H}). \quad (25)$$

## C EXTENSIVE EXPERIMENTAL RESULTS

We proceed to visually compare the performance of our H-EDML with conventional Euclidean GNNs (GCN as a representative method) and HGNNs (HGAT as a representative method), alongside a contrastive learning approach (HGCL as a representative method), as illustrated in Figure 6. The following observations emerge: firstly, HGAT demonstrates superior performance over GCN on datasets characterized by low $\delta$-hyperbolicity such as Disease and Airport datasets, while GCN excels over HGAT on citation datasets with high $\delta$-hyperbolicity. This distinction underscores the proficiency of HGNNs in capturing hierarchical structures compared to GNNs' adeptness at flat structures. Secondly, the amalgamation of Euclidean and hyperbolic spaces in H-EDML and HGCL results in superior performance compared to GCN and HGAT, emphasizing the efficacy of leveraging the strengths of both spaces. Lastly, our proposed H-EDML outperforms HGCL generally through deep mutual learning and probabilistic integration in decision-making processes.

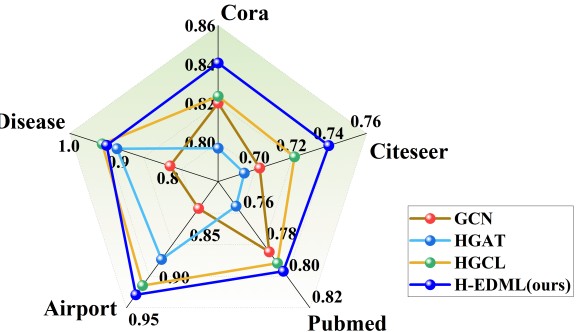

**Figure 6: Intuitive comparison of the performance of four classes of methods: the Euclidean GNNs method GCN, the hyperbolic GNNs method HGAT, the contrastive learning method HGCL, and our proposed method H-EDML.**

