# OpenReview forum: "Hyperbolic-Euclidean Deep Mutual Learning"
_ACM.org/TheWebConf/2025/Conference — WWW 2025 Poster_

### Official Review · Reviewer_7FTQ · 2024-11-26

**Novelty:** 5
**Technical Quality:** 5

**Review:**

The authors propose the **H-EDML framework**, which leverages the unique properties of hyperbolic space to effectively capture the hierarchical relationships present in graph data, while also utilizing the familiar Euclidean space to handle local interactions. They further enhance the model's performance with a topology mutual learning module.

## Pros
- The proposed method demonstrates consistent performance regardless of the hyperbolicity of the data.
- The use of the topology mutual learning module effectively improves the model’s performance.
- The authors conducted extensive experiments to validate their approach.

## Cons
- The datasets used in the experiments are too small to draw generalizable conclusions.
- The related work and experimental sections give limited consideration to contrastive learning methods combining GNNs and HGNNs.
- The paper focuses only on node classification, while graph classification is also an important task that should be considered.

**Questions:**

**Q1.** Regarding the datasets, the size of the data seems too small. For commonly used datasets like Cora, Citeseer, and Pubmed, it is unclear whether the model performs well due to its representational power or simply because overfitting is less likely to occur. Could the authors clarify this distinction?

**Q2.** In the related work section, contrastive learning methods that combine GNNs and HGNNs ([20], [26], [46]) are not fully discussed. In the experimental section, only [26] is used as a baseline. While [20] is related to recommendation and can be excluded, [46] appears to be a relevant baseline. Why was [46] not considered in the experiments?

**Reviewer Confidence:**

2: The reviewer is willing to defend the evaluation, but it is likely that the reviewer did not understand parts of the paper

**Scope:**

3: The work is somewhat relevant to the Web and to the track, and is of narrow interest to a sub-community

---

### Official Review · Reviewer_4qQP · 2024-11-27

**Novelty:** 5
**Technical Quality:** 5

**Review:**

The authors propose a Hyperbolic-Euclidean Deep Mutual Learning (H-EDML) framework for graph neural networks, combining hyperbolic and Euclidean geometries to leverage their respective strengths. This approach integrates a topology mutual learning module to improve the structural perception of each model and a decision mutual learning module for enhanced decision-making. An attention-based probabilistic integration strategy is also implemented to address disparities in decision-making, showing competitive performance on multiple datasets.

## Strengths:
1. The combination of hyperbolic and Euclidean spaces allows the model to capture both hierarchical and flat graph structures effectively.
2. The topology and decision mutual learning modules improve both structure perception and decision accuracy.
3. Experimental results demonstrate H-EDML’s robustness and competitive performance, outperforming state-of-the-art methods.

## Weaknesses:
1. The two-stage training approach and multiple modules may increase computational overhead.
2. Performance gains might be specific to graph datasets with hierarchical structures, limiting generalizability.
3. The method’s effectiveness relies on the quality of base Euclidean and hyperbolic GNN models.

**Questions:**

1)	Are all tasks suitable for a combination of hyperbolic and Euclidean geometries? If not, what types of tasks are more suitable?
2)	What does the “generalization” mean when integrating the decision mutual learning module? Which experimental results demonstrate the generalization?

**Reviewer Confidence:**

3: The reviewer is confident but not certain that the evaluation is correct

**Scope:**

3: The work is somewhat relevant to the Web and to the track, and is of narrow interest to a sub-community

---

### Official Review · Reviewer_LEDd · 2024-11-28

**Novelty:** 6
**Technical Quality:** 6

**Review:**

The authors propose a two-stage training framework called H-EDML which leverages hyperbolic space and Euclidean space. Their main idea is to achieve complementary information exchange through deep mutual learning between hyperbolic and Euclidean GNNs. Two modules are introduced in the proposed framework, which are topology mutual learning and decision mutual learning. These two modules extensively contribute to the model performance in the first stage training. Stage-2 mitigates any potential information redundancy or conflicts from the concurrent trainings. Experimental results show the efficacy of the proposed model.

- Strengths
1. The main idea of the paper is very interesting tackling the important problem in GNN community: trying to solve various types of graphs.
2. The proposed model is novel and creative. The authors have carefully designed the two-stage learning process addressing the challenges involved.
3. The experimental results showcases how effective the proposed model is. Their experimental analysis really help the readers understand how each module contribute to the achieve performances. Most of all, the reported results in Table 2 are impressive.

- Weaknesses
1. Only small datasets have been used in the experimentation.
2. More dataset with small $\rho$ (perhaps between 1 and 0 or between 1 and 3.5?) would better show how the proposed model can be generalized.

(others)
1.  While instead of while ? (line 223)
2.  $\delta$ is both used in Equation 18 and for computing hyperbolicity.

**Questions:**

1. Any thoughts why the separability between class 2 and class 3 has been significantly improved? If the authors can explain more what each label means it will be insightful.
2. Experiments are conducted on small datasets, how well does HGCL scale to large datasets?

**Reviewer Confidence:**

4: The reviewer is certain that the evaluation is correct and very familiar with the relevant literature

**Scope:**

4: The work is relevant to the Web and to the track, and is of broad interest to the community